# Unsupervised Adversarial Anomaly Detection using One-Class Support Vector Machines

## Abstract

Anomaly detection discovers regular patterns in unlabeled data and identifies the non-conforming data points, which in some cases are the result of malicious attacks by adversaries. Learners such as One-Class Support Vector Machines (OCSVMs) have been successfully used in anomaly detection, yet their performance may degrade significantly in the presence of sophisticated adversaries, who target the algorithm itself by compromising the integrity of the training data. With the rise in the use of machine learning in mission critical day-to-day activities where errors may have significant consequences, it is imperative that machine learning systems are made secure. To address this, we propose a defense mechanism that is based on a contraction of the data, and we test its effectiveness using OCSVMs. The proposed approach introduces a layer of uncertainty on top of the OCSVM learner, making it infeasible for the adversary to guess the specific configuration of the learner. We theoretically analyze the effects of adversarial perturbations on the separating margin of OCSVMs and provide empirical evidence on several benchmark datasets, which show that by carefully contracting the data in low dimensional spaces, we can successfully identify adversarial samples that would not have been identifiable in the original dimensional space. The numerical results show that the proposed method improves OCSVMs performance substantially (2-7%).

## 1 Introduction

Anomaly detection refers to the problem of discovering patterns in data and identifying data points that do not conform to the learned patterns. These non-conforming data points are often referred to as anomalies or outliers. Anomaly detection has numerous applications in a variety of domains such as network intrusion detection, credit card fraud detection, and spam filtering. It is an important problem since the presence of anomalies may indicate malicious attacks that could disrupt mission critical operations. Many machine learning methods, such as One-Class Support Vector Machines (OCSVM) (Schölkopf et al., 2000), have been proven to be effective in anomaly detection applications. Although they are designed to withstand the effects of random noise in data, when adversaries deliberately alter the input data and compromise their integrity, the performance of these learning algorithms may degrade significantly.

Anomaly detection systems are often deployed in environments where the data naturally evolves. In such situations, the models need to be retrained periodically, in contrast to many conventional machine learning applications, where the current and future data is assumed to have identical properties. This periodic training may allow adversaries to gradually inject malicious data to diminish the decision making capabilities of the learning algorithms (Huang et al., 2011). The aim of the adversaries may be to avoid the detection of attacks or to decrease the performance of the learning system (Huang et al., 2011). To achieve these aims, adversaries can undermine learning algorithms in several ways. For instance, they may manipulate the training data if it is gathered from the real operation of a system (e.g., spam filtering, firewall, anti-virus, etc.) and force the learning algorithm to learn a distorted representation that is favorable to them.

A sophisticated adversary has the capacity to conduct an attack in numerous ways. Hence, it is not feasible to provide a general analysis that covers the whole range of attacks, across different machine learning algorithms. In this work, we explore the following key question: Is it possible to make OCSVMs more resistant against adversarial attacks which target the integrity of the training data through distortions?. If an adversary can maliciously perturb the input data used by a learning algorithm, they can force the learner to learn a model that is favorable to them. It has become imperative to secure machine learning systems against such adversaries due to the recent increase of automation in many day to day applications. In the context of image recognition, the perturbations caused by an adversary are usually imperceptible to humans, but they can force a learned model to mis-classify the perturbed images with high confidence. As Evtimov et al. (2017) have shown, with the emergence of self driving vehicles, an adversary could alter a *"S-T-O-P"* road sign in such a way that a vehicle (learning system) would reliably classify it as a *"Speed Limit 45"* sign. Such perturbations could be imperceptible to humans and could result in the loss of human lives.

Our goal is to utilize a nonlinear data projection based algorithm to increase the attack resistance of OCSVMs against an adversarial opponent under realistic assumptions. The theory of nonlinear random projections facilitates large-scale, data-oriented, multi-agent decisions by reducing the number of optimization parameters and variables. Recent work in the literature shows that nonlinear random projections improve the training and evaluation times of kernel machines, without significantly compromising the accuracy of the trained models (Rahimi & Recht, 2008; Erfani et al., 2015). In this paper, we show that under adversarial conditions, selective nonlinear random projections can be leveraged to increase the attack resistance of OCSVMs as well.

A dataset $X \in \mathbb{R}^{n \times d}$ that is projected using a carefully chosen projection matrix $A \in \mathbb{R}^{d \times r}$ comprised of random elements that are normally distributed, would have its pairwise Euclidean distances preserved with high probability in the projected space $XA$ (Johnson & Lindenstrauss, 1984). Therefore, the properties of the original data distribution would be present in the projected dataset with only minor perturbations. Note that here $r$ is the dimension to which the data is nonlinearly projected and $r < d$. Since the elements of $A$ are drawn randomly, the learner obtains an additional layer of security as it becomes virtually impossible for the adversary to guess the projection mechanism used by the learner due to the search space becoming unbounded.

More formally, let $\|w_{pd}^*\|_2$ be the length of the weight vector of the OCSVM in the transformed space, after solving the corresponding optimization problem that includes the distortion made by the adversary and the nonlinear random projection. Let $\|w_p^*\|_2$ be the length of the weight vector in the transformed space, where there is no adversary present. Since the learner cannot distinguish between the original data and the distorted data, the learner would not have the ability to explicitly calculate $\|w_p^*\|_2$. Therefore, for reasonable values of $r$ and small distortions $D$, we prove in this paper that $\|w_p^*\|_2$ is bounded above:

$$\left\| w_p^* \right\|_2 \le (1 + \epsilon) \left\| w_{pd}^* \right\|_2. \tag{1}$$

The main **contributions** of this work are summarized as follows. We derive analytically an upper bound on the length of the weight vector of a OCSVM trained on an undistorted dataset that has been nonlinearly transformed to a lower dimensional space. In addition, the resistance added by nonlinear data transformations against an adversarial opponent is studied through numerical experiments on several benchmark datasets. We believe that our proposed approach can (i) increase the attack resistance of OCSVMs under adversarial conditions, and (ii) give the learner a significant advantage from a security perspective by adding a layer of unpredictability through the randomness of the data transformation in a selective direction.

## 2 BACKGROUND AND RELATED WORK

As our proposed approach on adversarial learning for anomaly detection is based on randomized kernels, in this section we briefly review these two lines of research.

## 2.1 Randomized Kernels for SVMs

To improve the efficiency of kernel machines, Rahimi & Recht (2008) embedded a random projection into the kernel formulation. They introduced a novel, data independent method (Random Kitchen Sinks (RKS)) that approximates a kernel function by mapping the dataset to a relatively low dimensional randomized feature space. Instead of relying on the implicit transformation provided by the kernel trick, Rahimi and Recht explicitly mapped the data to a low-dimensional Euclidean inner product space using a randomized feature map $z : \mathbb{R}^d \to \mathbb{R}^r$. The kernel value of two data points is then approximated by the dot product between their corresponding points in the transformed space $z$. As $z$ is a low dimensional transformation, compared to basis expansion, it is more computationally efficient to transform inputs with $z$ and train a linear SVM, as the result is comparable to that of its corresponding nonlinear SVM. Evaluating a new data-point also becomes efficient when randomized feature maps are used. Subsequently, Le et al. (2013) introduced a transformation method that has lower time and space complexities compared to RKS.

More recently, the method of Rahimi & Recht (2008) has been applied to other types of kernel machines. Erfani et al. (2015) introduced *Randomized One-class SVMs (R1SVM)*, an unsupervised anomaly detection technique that uses randomized, nonlinear features in conjunction with a linear kernel. They reported that R1SVM reduces the training and evaluation times of OCSVMs by up to two orders of magnitude without compromising the accuracy of the predictor. Our work differs from these as we look at random projections as a defense mechanism for OCSVMs under adversarial conditions. However, to the best of our knowledge, no existing work adopts Rahimi and Recht's method to address adversarial learning for anomaly detection with OCSVMs.

## 2.2 Learning under adversarial conditions

The problem of adversarial learning has inspired a wide range of research from the machine learning community, see Barreno et al. (2010) for a survey. For example, Zhou et al. (2012) introduced an Adversarial Support Vector Machine (AD-SVM) model. AD-SVM incorporated additional constraint conditions to the binary SVM optimization problem in order to thwart an adversary's attacks. Their model leads to unsatisfactory results when the severity of real attacks differ from the expected attack severity by the model. While we gain valuable insights regarding attack strategies from this work, the defense mechanism in our work is significantly different. Furthermore, our work primarily focuses on unsupervised learning, whereas Zhou et al. (2012) use a binary SVM for their work.

Deep Neural Networks (DNNs) have been shown to be robust to noise in the input data (Fawzi et al., 2016), but they are unable to withstand carefully crafted adversarial data points (Goodfellow et al., 2014). While these works are in the same domain, they are not directly related to our work, which uses OCSVMs and kernels. Recent work by Evtimov et al. (2017) showed that an attacker could alter a *"S-T-O-P"* road sign in such a way that a vehicle (learning system) would reliably classify it as a *"Speed Limit 45"* sign. Such perturbations could be imperceptible to humans and could result in the loss of human lives.

This paper presents a unique framework that brings together adversarial learning, anomaly detection using OCSVMs, and randomized kernels. To the best of our knowledge, no existing work has explored this unique path.

## 3 Problem Definition and Attack Model

This section presents the problem definition and the interaction between the adversary and the learner.

## 3.1 Problem Definition

We consider an adversarial learning problem for anomaly detection in the presence of a malicious adversary. The adversary modifies the training data in order to disrupt the learning process of the learner, who aims to detect anomalous data points. Hence, the adversary's main goal is to hinder the decision making capability of the learning system by compromising the integrity of the input data. Let $X \in \mathbb{R}^{n \times d}$ be the training dataset that contains data from the normal class and $D \in \mathbb{R}^{n \times d}$ be the

perturbations made by the adversary, making $X + D$ the training dataset that has been distorted. It should be noted that the learner cannot demarcate $D$ from $(X + D)$, otherwise the learner would be able to remove the adversarial distortions during training, making the problem trivial. The adversary has the freedom to determine $D$ based on the knowledge it possesses regarding the learning system, although the magnitude of $D$ is usually bounded due to its limited knowledge, the increased risk of being discovered, and computational constraints.

The learner, in response to the adversary's perturbations, projects the data to a lower dimensional space (i.e., $(X + D)A$). Each sample $(X + D)_i$ is then non-linearly transformed using the function $z((X + D)_i) = \sqrt{2}\cos\left((X + D)_i A + b\right)$, where $b$ is a $r$-dimensional vector whose elements are drawn uniformly from $[0, 2\pi]$. This nonlinear transformation to a lower dimensional space is done in order to minimize the effectiveness of the attacks. The specifics of how to select a good transformation will be explained in Section 3.3. Previously it has been shown that by transforming the data using $z$, we can approximate nonlinear kernels such as the Radial Basis Function (RBF), thereby reducing the computational and memory overheads that impede kernel based learning algorithms (Rahimi & Recht, 2008).

The anomaly detection problem is addressed in this paper using the OCSVM algorithm introduced by Schölkopf et al. (2000), which separates the training data from the origin with a maximal margin in the transformed space. Remember that $(X + D) \in \mathbb{R}^{n \times d}$ is the original training dataset and let $C \in \mathbb{R}^{n \times r}$ be the result of the nonlinear transformation $z$. As explained in Section 2.1, since $C$ can now be linearly separated, the dual form of the OCSVM algorithm can be written in matrix notation as,

$$
\begin{aligned}
\underset{\alpha}{\text{minimize}} \quad & \frac{1}{2}\alpha^T CC^T \alpha, \\
\text{subject to} \quad & 0 \leq \alpha \leq \frac{1}{\nu n}, \\
& \mathbf{1}^T \alpha = 1,
\end{aligned}
\tag{2}
$$

where $\alpha$ is the vector of Lagrange multipliers, $\nu \in (0, 1]$ is a parameter that defines an upper bound on the fraction of support vectors and a lower bound on the fraction of outliers, and $\mathbf{1}$ is a vector of ones. The margin of the optimal separating hyperplane is given by $\rho/\|w\|_2$, where $w = \alpha^T C$ ($\alpha$ is the solution to (2)). The offset of the hyperplane (i.e., $\rho$) can be recovered using any of the support vectors with $0 < \alpha_i < 1/\nu n$.

## 3.2 ATTACK MODEL

In an *integrity attack*, the adversary desires false negatives, and hence, would use $D$ to move the separating hyperplane of the OCSVM away from the normal data cloud and towards the space where anomalies lie. Whereas in an *availability attack*, the adversary would attempt to create false positives by shifting the separating margin towards the normal data cloud (Huang et al., 2011). This work focuses on integrity attacks, where the objective of the adversary is to minimize the margin of separation by maliciously injecting data into the training dataset through $D$. We will focus on targeted attacks, where the adversary tries to smuggle a particular class of anomalies across the separating margin. The adversary has the assumed capability to move any data point in any direction by adding a non-zero displacement vector $\kappa_i$ to $x_i$. It is also assumed that the adversary does not have any knowledge about the projection mechanisms used by the learner. Therefore, all of the adversary's actions take place in the original full dimensional space.

The attack model used is inspired by the restrained attack model described by Zhou et al. (2012). The adversary would select a random subset of anomalies, push them towards the normal data cloud and inject these perturbed points into the training set. Since the OCSVM algorithm considers all the data points in the training set to be from the normal class, these distorted anomalies would be seen by the learning algorithm as normal data points (similar to label flipping). The severity of the attack, controlled by the parameter $s_{attack} \in [0, 1]$, is proportional to the distance from the normal data cloud. To clarify, an anomaly data point that is pushed closer to the normal data cloud (small $s_{attack}$) would be considered as a *moderate attack*, whereas, an anomaly data point that is farther away from the normal data (large $s_{attack}$) cloud would be considered a *severe attack*. As Figure 1 shows, when a digit '7' is distorted with less severity (e.g., 0.2), it resembles a '9' visually. But as the attack severity increases, the digit tends to look like a '7' even though it is labeled as a '9'.

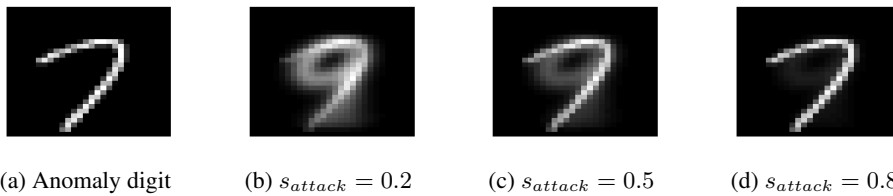

(a) Anomaly digit     (b) $s_{attack} = 0.2$     (c) $s_{attack} = 0.5$     (d) $s_{attack} = 0.8$

Figure 1: A digit from anomaly class ('7') distorted to appear like a digit from the normal class ('9')

The adversary picks a target $x_i^t$ for each $x_i$ to be distorted and moves it towards the target by some amount. Choosing $x_i^t$ for each $x_i$ optimally requires a significant level of computational effort and a thorough knowledge about the distribution of the data. The attacker, similar to Zhou et al. (2012), uses the centroid of the normal data cloud in the training set as the target point for all anomaly data points that he/she intends to distort. A data point sampled from the normal class or an artificial data point generated from the estimated normal class distribution could be used as alternatives.

For each attribute $j$ in the original feature space, the adversary is able to add $\kappa_{ij}$ to $x_{ij}$, where

$$\kappa_{ij} = (1 - s_{attack})(x_{ij}^t - x_{ij}), \tag{3}$$

$$|\kappa_{ij}| \leq |x_{ij}^t - x_{ij}|, \forall j \in d. \tag{4}$$

The adversary is able to orchestrate different attacks by changing the percentage of distorted anomaly data points in the training dataset (i.e., $p_{attack}$) and the severity of the distortion (i.e., $s_{attack}$). It should be noted that if the adversary greedily pushes data away from the normal data cloud, it would result in the distortions becoming quite evident and increase the risk of discovery of the attack. Figure 2 illustrates the data distributions when different levels of attack severities are applied to the anomaly data. As $s_{attack}$ increases, the anomaly data points are moved farther away from the normal data cloud, altering the position of the separating hyperplane.

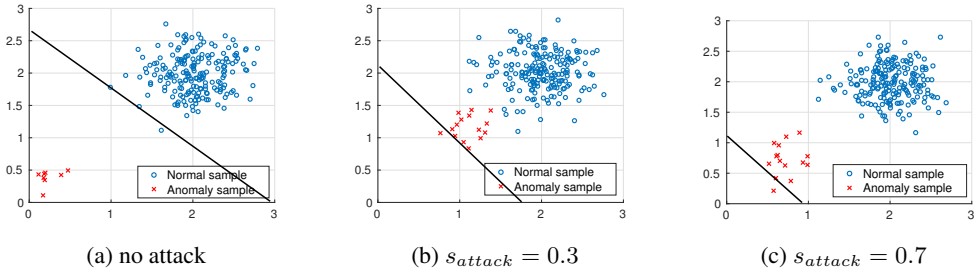

(a) no attack     (b) $s_{attack} = 0.3$     (c) $s_{attack} = 0.7$

Figure 2: Training data distribution and separating hyperplane of a toy problem under different attack severities. 'o' denotes the undistorted data points and 'x' denotes the data points distorted by the adversary. The OCSVM is trained using the entire (unlabeled) dataset as normal.

### 3.3 DEFENSE STRATEGY

Anticipating possible distortions by an adversary, the learner can take precautions to minimize their effects by contracting the data to a lower dimensional space. By using a projection matrix $A$ with its elements drawn randomly from some distribution, the learner introduces a layer of uncertainty to the adversary-learner problem. This gives the learner an additional advantage from a security perspective. But this unpredictability can also be seen as the main caveat of using random projections to reduce the dimensionality of data. While some random projections result in better separated volumetric clouds than the original ones, some projections result in the data from different classes being overlapped.

In order to increase the attack resistance of a learning system, the impact of adversarial inputs should be minimized. Based on this intuition, we propose that a projection that conceals the potential

distortions of an adversary would make any learning system that learns from the projected data more resistant to attacks. As the learner cannot demarcate $D$ from the training data, it is not possible to identify an ideal projection that conceals the adversarial distortions. Thus, the learner would have to select a projection that contracts the entire training set (expecting the adversarial points to be masked by normal data) and separates the training data from the origin with the largest margin in the transformed space.

Therefore, motivated by a generalized version of the Dunn's index (Bezdek & Pal, 1998), we propose a compactness measure to identify suitable projection directions in a one-class problem. The learner would calculate the compactness of the projected data using Equation 5 for multiple random projections of the training data and select the one that gives the highest value, which can be considered as the projection that gives the best attack resistance.

The compactness of projection $P_i$, where $\mu_i$ is the centroid of the projected training set, $0$ is the origin in the transformed space and the function $d$ is the Euclidean distance can be calculated as,

$$\text{compactness of } P_i = \frac{d(0, \mu_i)}{\left(\sum_{x \in P_i} d(x, \mu_i)\right)/n}. \tag{5}$$

Intuitively, an anomaly detection algorithm would attempt to identify the smallest hypersphere that contains the training data set, in either the original dimensional space (i.e., $d$) or in a transformed space. The objective of the adversary (learner) in such a situation would be to maximize (minimize) the radius of the minimum enclosing hypersphere. The approach used by the learner to minimize the attack's effects is formalized next in terms of the random projection parameters $A$ and $b$, the dimension of the projected dataset $r$ and the adversary's data distortion strategy $D$.

---

**Algorithm 1** Defense mechanism

1: **procedure** IDENTIFY_PROJECTION($X + D$, $X_{test}$,r)
2:     $max\_compactness \leftarrow -1$
3:     $best\_transformation \leftarrow null$
4:     $N \leftarrow 1,000$                                  ▷ Number of projection directions to sample
5:     $A, b \leftarrow null$                                           ▷ Transformation parameters
6:     **for** $i \leftarrow 1, N$ **do**
7:         $[(X + D)^*, A, b] \leftarrow z(X + D)$             ▷ Nonlinearly transform the data
8:         $compactness \leftarrow calculate\_compactness((X + D)^*)$    ▷ Calculate compactness of transformed data. (Equation 5)
9:         **if** $compactness > max\_compactness$ **then**                 ▷ Identify best projection
10:             $max\_compactness \leftarrow compactness$
11:             $best\_transformation \leftarrow (X + D)^*$
12:             $A \leftarrow A$
13:             $b \leftarrow b$
14:         **end if**
15:     **end for**
16:     $model \leftarrow ocsvm\_train(best\_transformation)$ ▷ Train linear OCSVM in projected space
17:     $X_{test}^* \leftarrow z(X_{test}, A, b)$                      ▷ Transform the test set with same parameters
18:     $accuracy\_metrics \leftarrow svm\_predict(X_{test}^*, model)$      ▷ Evaluate the transformed test set
19:     **return** $A, b$   ▷ Return transformation parameters that result in the best defensive projection
20: **end procedure**

---

## 4   OCSVM UNDER AN ATTACK ON INTEGRITY

This section analyzes the effects of the adversary's perturbations on the margin of separation of the OCSVM. The distance between the hyperplane and the origin of a OCSVM is given by $\rho/\|w\|_2$, where $\rho$ is the offset and $w$ is the vector of weights. This implies that a small $\|w\|_2$ corresponds to a large margin of separation from the origin. Since the learner cannot demarcate the perturbations from the normal training data, it cannot empirically calculate this value for the undistorted dataset. Therefore, based on the assumptions given below, we analytically derive an upper bound on $\|w\|_2$

of a OCSVM that has been trained on a nonlinearly transformed dataset without any adversarial distortions.

**Assumption 1.** The distortions made by the adversary are small and positive.
This assumption holds well due to the following reasons,

1. The adversary distorts data in the original feature space. In this space, we can align any given dataset in such a way that any outliers present in the data would lie closer to the origin and the normal data cloud would lie in the positive orthant. Such a transformation would compel the adversary to make adversarial distortions in the direction of the normal data cloud (positive).

2. Large distortions increase the risk of the adversary being discovered, therefore a rational adversary would refrain from conducting attacks with significant perturbations.

**Theorem 1.** Let $\omega_p^*$ be the primal solution the OCSVM optimization problem in the projected space when there are no adversarial distortions. Similarly, define $\omega_{pd}^*$ as the primal solution in the presence of a malicious adversary. Let $r$ be the number of dimensions to which the data is projected. Then, for small, positive distortions, the length of the weight vector $w_p^*$ is bounded above,

$$\left\| w_p^* \right\|_2 \leq \left\| w_{pd}^* \right\|_2 + \epsilon + \sqrt{r}. \tag{6}$$

The strength of the adversary's attacks will be reflected on the value of $\epsilon$ and will increase with the strength of the attacks. The defender is able to make the upper bound of $\left\| w_p^* \right\|_2$ tighter by reducing the dimensionality of the dataset (i.e., $r$). Refer Table 2 for empirical validation and Appendix A for a proof.

## 5 EXPERIMENTS AND DISCUSSION

The experimental evaluation presented in the following section demonstrates the effectiveness of our proposed defense mechanism on three benchmark datasets: MNIST, CIFAR-10, and SVHN. We compare the performance of OCSVMs in conjunction with nonlinear random projections, when an active adversary is conducting a directed attack by maliciously distorting the data.

**Datasets:** We generate single-class (unlabeled) datasets from MNIST (LeCun & Cortes, 2010), CIFAR-10 (Krizhevsky, 2009) and SVHN (Yuval Netzer, 2011) by considering one of the original classes as the normal class, and another class in the dataset as the anomaly class. The objective of the adversary is to get the learner to classify anomalies as normal data points during evaluation. Note again that the combined training sets are single-class and unlabeled, while the original classes provide the ground truth.

For each dataset, we create two test sets (with a normal to anomaly ratio of $5 : 1$): (i) a clean test set (called test$_C$) that consists of data from the anomaly class and normal class, without undergoing any transformations; (ii) a distorted test set (test$_D$) with its anomalies pushed closer to normal data cloud similar to how distortions are made during the training phase. All values in the datasets are normalized by dividing by 255. Refer to Table 1 for details regarding the class and number of samples used in each training and test set.

Table 1: Datasets used for training and testing purposes.

| Dataset | Training set size | Test set size | normal class | anomaly class |
|---------|------------------:|--------------:|-------------:|--------------:|
| MNIST | 2,000 | 1,200 | digit '9' | digit '7' |
| CIFAR-10 | 3,650 | 1,200 | airplane | frog |
| SVHN | 4,200 | 1,200 | digit '0' | digit '1' |

**Experimental setup:** We use the OCSVM implementation of the LIBSVM library (Chang & Lin, 2011) in our experiments. Different attack scenarios are simulated (creating train$_D$) by varying

the attack percentage $p_{attack}$ and attack severity $s_{attack}$. We specifically choose the values 5% for $p_{attack}$ and 0.3, 0.5 and 0.7 for $s_{attack}$. For comparison purposes, we test all the attack scenarios with a OCSVM using the RBF kernel in the original feature space.

For nonlinear projections, we choose the dimensions to which the data is projected by selecting the local intrinsic dimensionality (LID) of the dataset (Amsaleg et al., 2015), and 50% of the original number of dimensions. The corresponding LID values are 220, 733 and 463 for MNIST, CIFAR-10 and SVHN respectively. For each dimension, the learner would perform 1,000 nonlinear transformations and select the projection that results in the highest compactness, using Equation 5. The test sets would undergo the same transformation as the selected one. The learner then uses the transformed training set to train a OCSVM with a linear kernel, and the resulting model is evaluated using the test sets. For these experiments the $\nu$ parameter of the OCSVM is kept fixed across all experiments conducted using each dataset. Since $\nu$ sets a lower bound on the fraction of outliers, it is crucial to keep its value fixed across different attack scenarios in order to evaluate the interplay between the adversarial distortions and the performance of the OCSVMs. When training the full dataset with the RBF kernel, the default value of the parameter *gamma* in LIBSVM was kept unchanged.

**Accuracy metric:** For comparison purposes, we also train a OCSVM using an undistorted training set (called train$_C$). We report the performance against test$_C$ as well as test$_D$ using the f-score. We observed similar patterns for each dataset across different experiments, but due to space limitations, graphs and tables of only some are shown.

## 5.1 RESULTS AND DISCUSSION

Figure 3 presents the results of an experiment, where $p_{attack} = 5\%$ and $s_{attack} = 0.5$ (refer to Appendix B for the corresponding numerical values). The top row shows how the f-score is affected by the non-linear transformation and the adversary's distortion. For each number of dimensions, four results are presented; f-score when: (i) trained using train$_C$, and tested with test$_C$; (ii) trained with train$_C$ and tested with test$_D$; (iii) trained with train$_D$ and tested with test$_C$; and finally (iv) trained with train$_D$ and tested with test$_D$.

First, the classification performance of OCSVMs trained on nonlinearly transformed data are 2-7% higher than the performance of the OCSVM trained on the original feature space, although they require far less computation time (e.g., for CIFAR-10, a training time of 4.95s when trained with $r = 733$ vs. 20.24s when trained with the full, $3,072$ dimesional data). These observations are in line with the previous work in this area (Erfani et al., 2015). Therefore, the range of the $y$ axes in the graphs have been altered so that the differences can be observed.

We observe that the f-scores across the dimensions decrease between train$_C$|test$_D$ and train$_D$|test$_D$. This indicates that a OCSVM trained on clean data can identify adversarial samples better than a OCSVM trained on distorted data. Consequently this shows that OCSVMs are not immune to integrity attacks by design, and by carefully crafting adversarial data points, adversaries can manipulate OCSVMs to learn models that are favorable to them.

A comparison between f-scores of train$_D$|test$_C$ and train$_D$|test$_D$ shows that, as the dimension is reduced from the original dimension, the f-scores increase, but as we reduce the dimension further, the f-scores begin to decrease. The increase in f-score confirms that by projecting data to a lower dimensional space using a carefully selected direction, we can identify adversarial samples that would not have been identifiable in the original feature space. This is confirmed by the graphs in the second row, which show the false positive rate of the OCSVMs under integrity attacks (i.e., number of anomalies that are undetected). We find that there is a significant improvement in detecting adversarial samples under the proposed approach (e.g., 23% on CIFAR-10 and 31% on MNIST).

When the dimensions are reduced below a certain dataset dependent threshold the OCSVM performance starts to decline (e.g., SVHN 1,500 vs 463). We postulate that the explanation of this effect is the reduction in distance between classes (in this case perturbed anomalies and normal data points) with the dimension. As we reduce the dimension of the transformation, we are able to reduce the effects of the adversarial datapoints. But at the same time, there is a significant loss of useful information due to the dimensionality reduction. Due to the interplay between these two factors, the performance of OCSVMs reduces as we decrease the dimension beyond a certain threshold.

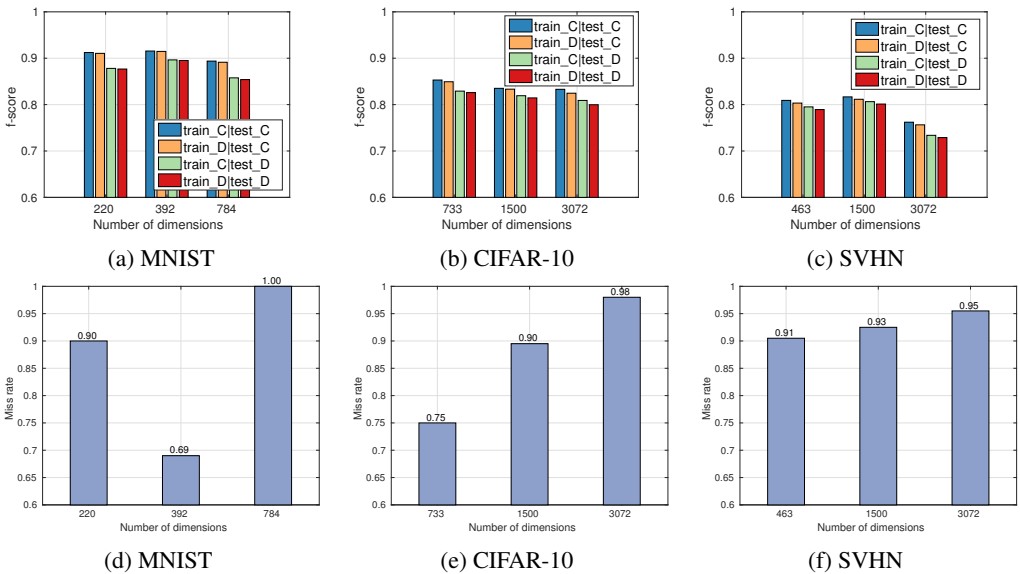

Figure 3: The top row shows the performance of OCSVMs under adversarial conditions when the training takes place in different dimensional spaces. It compares the evaluation performance of OCSVMs trained on $\text{train}_C$ and $\text{train}_D$ against the two test sets: $\text{test}_c$ and $\text{test}_D$. The bottom row shows the false positive rate of the OCSVMs under an integrity attack (i.e., trained on $\text{train}_D$ and evaluated using $\text{test}_D$).

Table 2: Comparison of actual $\left\|w_p^*\right\|_2$, calculated on the MNIST data set ($p_{attack} = 5\%$ and $s_attack = 0.5$) and the theoretical upperbound calculated using Theorem 1.

| dimensions | Actual $\left\|w_p^*\right\|_2$ | $\left\|w_{pd}^*\right\|_2$ | $\sqrt{r}$ | Upperbound with $\epsilon = 0$ |
|---|---|---|---|---|
| 210 | 1,969.30 | 2,073.00 | 14.49 | 2,087.49 |
| 393 | 2,734.40 | 2,878.30 | 19.82 | 2,898.12 |

Finally, Table 2 shows the effectiveness of the bound derived in Theorem 1. The results show the consistency of the upper bound, which becomes tighter under dimension reduction.

In summary, the above experiments demonstrate that, (i) OCSVMs are vulnerable to adversarial *attacks on integrity*, (ii) by projecting a distorted dataset to a lower dimension in an appropriate direction we can increase the robustness of the learned model w.r.t. integrity attacks, (iii) the performance, in terms of f-score, starts to decline when the dimensionality is reduced beyond a certain threshold, and (iv) the performance in the projected spaces, when there are no attacks on integrity, is comparable to that in the original dimensional space, but with less computational burden.

## 6 CONCLUSIONS AND FUTURE WORK

This paper presents a theoretical and experimental investigation based on a unique combination of unsupervised anomaly detection, using OCSVMs and random projections for dimensionality reduction in the presence of a sophisticated adversary. Our numerical analysis focuses on two main aspects: the performance of OCSVMs in lower dimensional spaces under adversarial conditions and the impact of nonlinear random projections on the robustness of OCSVMs w.r.t. adversarial perturbations. The results suggest that OCSVMs can be significantly affected if an adversary has access to the data on which they are trained. For each dataset, with very high probability, there is at least one dimensionality and projection direction that results in a OCSVM that is able to identify adversarial samples that would not have been identifiable by a OCSVM in the original dimensional space. Due

to the layer of uncertainty added by the randomness of the projection, our approach makes the learning system more secure by making it virtually impossible for an adversary to guess the underlying details of the learner. Therefore, our approach can be utilized to make a learning system secure by, (i) reducing the impact of possible adversarial perturbations by contracting, and moving the normal data cloud away from the origin in the projected space, and (ii) making the search space of the adversary unbounded by adding a layer of randomness.

Since data contraction is at the core of our proposed approach, for our future work we would like to investigate whether our approach will still hold if used with other learning algorithms. One major question that arises from this work is how to optimally select the number of dimensions to transform the data to. We are currently exploring the possibility of using the intrinsic dimensionality of datasets to address this problem. Since there is a clear information asymmetry between the adversary and learner (due to the randomness), this problem provides a good foundation to explore game-theoretical formulations of anomaly detection and adversarial learning problems under dimensionality reduction techniques. We also plan to study "boiling frog" type of attacks, where the adversary gradually injects malicious data over time.

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

## A  PROOFS

**Definition 1.** Let $X \in \mathbb{R}^{n \times d}$ be the matrix that contains the training data. Similarly, define $D \in \mathbb{R}^{n \times d}$ as the matrix that contains the distortions made by the Adversary. Let $A \in \mathbb{R}^{d \times r}$ be the projection matrix where each element is an i.i.d. $\mathcal{N}(0, 1)$ random variable. Define $b$ as a $1 \times r$ row vector where each element is drawn uniformly from $[0, 2\pi]$. Using these variables, we define $C \in \mathbb{R}^{n \times r}$, where the element at row $i$ column $j$ takes the following form.

$$C_{i,j} = \cos\left(\left[\left(X_{i,1} + D_{i,1}\right)A_{1,j} + \left(X_{i,2} + D_{i,2}\right)A_{2,j} + \cdots + \left(X_{i,d} + D_{i,d}\right)A_{d,j}\right] + b_{1,j}\right), \quad (7)$$

$$\begin{aligned} C_{i,j} = \cos\Big(&\left(\left[X_{i,1}A_{1,j} + X_{i,2}A_{2,j} + \cdots + X_{i,d}A_{d,j}\right] + b_{1,j}\right) \\ &+ \left[D_{i,1}A_{1,j} + D_{i,2}A_{2,j} + \cdots + D_{i,d}A_{d,j}\right]\Big). \end{aligned} \quad (8)$$

Similarly, we define the matrices $C^X, C^D, S^X, S^D$ as follows,

$$\begin{aligned} C_{i,j}^X &= \cos\left(\left[X_{i,1}A_{1,j} + X_{i,2}A_{2,j} + \cdots + X_{i,d}A_{d,j}\right] + b_{1,j}\right), \\ C_{i,j}^D &= \cos\left(\left[D_{i,1}A_{1,j} + D_{i,2}A_{2,j} + \cdots + D_{i,d}A_{d,j}\right]\right), \\ S_{i,j}^X &= \sin\left(\left[X_{i,1}A_{1,j} + X_{i,2}A_{2,j} + \cdots + X_{i,d}A_{d,j}\right] + b_{1,j}\right), \\ S_{i,j}^D &= \sin\left(\left[D_{i,1}A_{1,j} + D_{i,2}A_{2,j} + \cdots + D_{i,d}A_{d,j}\right]\right). \end{aligned}$$

**Proof: (of Theorem 1)** Let $\tilde{\alpha}$ be the vector achieving the optimal solution in the projected space when adversarial distortions are present. Then, the solution for the primal problem in the projected space with adversarial distortions, defined as $w_{pd}^*$ can be obtained as

$$\left\|w_{pd}^*\right\|_2 = \left\|\tilde{\alpha}^T C\right\|_2. \quad (9)$$

Using the Cosine angle-sum identity on the matrix defined by equation 8 (the symbol $\odot$ denotes the Hadamard product for matrices),

$$\left\|w_{pd}^*\right\|_2 = \left\|\tilde{\alpha}^T\left(C^X \odot C^D\right) - \tilde{\alpha}^T\left(S^X \odot S^D\right)\right\|_2. \quad (10)$$

Using the reverse triangle inequality we obtain

$$\left\|w_{pd}^*\right\|_2 \geq \left\|\tilde{\alpha}^T\left(C^X \odot C^D\right)\right\|_2 - \left\|\tilde{\alpha}^T\left(S^X \odot S^D\right)\right\|_2. \quad (11)$$

From the constraint conditions of the OCSVM problem (refer equation 2), we get $\mathbf{1}^T\tilde{\alpha} = 1$. Also, as $\sin(\theta) \in [-1, 1], \forall \theta$. Therefore the inequality can be further simplified as,

$$\left\|w_{pd}^*\right\|_2 \geq \left\|\tilde{\alpha}^T\left(C^X \odot C^D\right)\right\|_2 - \sqrt{r}. \quad (12)$$

Due to *Assumption 1*, using small-angle approximation, for small $\epsilon$ values, we obtain

$$\left\|w_{pd}^*\right\|_2 \geq \left\|\tilde{\alpha}^T C^X\right\|_2 - \epsilon - \sqrt{r}. \quad (13)$$

Since the optimization problem is a minimization problem, as shown in (2), the optimal solution for the OCSVM without any distortion (i.e., $\alpha^*$) would give a value less than or equal to the value given by $\tilde{\alpha}$. Thus,

$$\left\| \alpha^{*,T} C^X \right\|_2 \leq \left\| w_{pd}^* \right\|_2 + \epsilon + \sqrt{r}. \tag{14}$$

Define $w_p^*$ as the primal solution optimization in the projected space, if there were no adversarial perturbations present, therefore

$$\left\| w_p^* \right\|_2 \leq \left\| w_{pd}^* \right\|_2 + \epsilon + \sqrt{r}. \tag{15}$$

## B  RESULTS

Table 3: Comparison of f-score on the distorted and undistorted MNIST test sets when $p_{attack}$ is set to 5% and $s_a ttack$ is 0.5

| dimensions | fp-rate | \multicolumn{4}{c}{f-score} |
| | | $train_C\|test_C$ | $train_C\|test_D$ | $train_D\|test_C$ | $train_D\|test_D$ |
| --- | --- | --- | --- | --- | --- |
| 220 | 0.900 | 0.9123 | 0.8781 | 0.9105 | 0.8767 |
| 392 | 0.690 | 0.9155 | 0.8964 | 0.9147 | 0.8951 |
| 784 | 1.000 | 0.8938 | 0.8577 | 0.8913 | 0.8539 |

Table 4: Comparison of f-score on the distorted and undistorted CIFAR-10 test sets when $p_{attack}$ is set to 5% and $s_a ttack$ is 0.5

| dimensions | fp-rate | $train_C\|test_C$ | $train_C\|test_D$ | $train_D\|test_C$ | $train_D\|test_D$ |
| --- | --- | --- | --- | --- | --- |
| 733 | 0.750 | 0.8530 | 0.8291 | 0.8493 | 0.8259 |
| 1500 | 0.895 | 0.8351 | 0.8192 | 0.8333 | 0.8145 |
| 3072 | 0.980 | 0.8330 | 0.8090 | 0.8246 | 0.7998 |

Table 5: Comparison of f-score on the distorted and undistorted SVHN test sets when $p_{attack}$ is set to 5% and $s_{attack}$ is 0.5

| dimensions | fp-rate | $train_C\|test_C$ | $train_C\|test_D$ | $train_D\|test_C$ | $train_D\|test_D$ |
| --- | --- | --- | --- | --- | --- |
| 463 | 0.8091 | 0.7951 | 0.8033 | 0.7893 | 0.9050 |
| 1500 | 0.8167 | 0.8064 | 0.8115 | 0.8012 | 0.9250 |
| 3072 | 0.7621 | 0.7338 | 0.7564 | 0.7289 | 0.9550 |

