# OpenReview forum: "Unsupervised Adversarial Anomaly  Detection using One-Class Support Vector Machines"
_ICLR.cc/2018/Conference — Reject_

### Official Review · AnonReviewer3 · 2017-11-27
**Empirically promising, but lacks in motivation**

**Rating:** 4
**Confidence:** 4

**Review:**

The authors propose a defense against attacks on the security of one-class SVM based anonaly detectors. The core idea is to perform a random projection of the data (which is supposed to decrease the impact from adversarial distortions). The approach is empirically tested on the following data: MNIST, CIFAR, and SVHN.

The paper is moderately well written and structured. Command of related work is ok, but some relevant refs are missing (e.g., Kloft and Laskov, JMLR 2012). The empirical results actually confirm that indeed the strategy of reducing the dimensionality using random projections reduces the impact from adversarial distortions. This is encouraging. What the paper really lacks in my opinion is a closer analysis of *why* the proposed approach works, i.e., a qualitative empirical analysis (toy experiment?) or theoretical justification. Right now, there is no theoretical justification for the approach, nor even a (in my opinion) convincing movitation/Intuition behind the approach. Also, the attack model should formally introduced.

In summary, I d like to encourage the authors to further investigate into their approach, but I am not convinced by the manuscript in the current form. It lacks both in sound theoretical justification and intuitive motivation of the approach. The experiments, however, show clearly advantages of the approach (again, here further experiments are necessary, e.g., varying the dose of adversarial points).

---

### Official Review · AnonReviewer1 · 2017-11-29
**Paper needs a clearer explanation and more thorough experiments**

**Rating:** 4
**Confidence:** 4

**Review:**

In this paper, the authors explore how using random projections can be used to make OCSVM robust to adversarially perturbed training data.  While the intuition is nice and interesting, the paper is not very clear in describing the attack and the experiments do not appropriately test whether this method actually provides robustness.

Details:
have been successfully in anomaly detection --> have been successfully used in anomaly detection

"The adversary would select a random subset of anomalies, push them towards the normal data cloud and inject these perturbed points into the training set" -- This seems backwards.  As in the example that follows, if the adversary wants to make anomalies seem normal at test time, it should move normal points outward from the normal point cloud (eg making a 9 look like a weird 7).

As s_attack increases, the anomaly data points are moved farther away from the normal data cloud, altering the position of the separating hyperplane.  -- This seems backwards from Fig 2.  From (a) to (b) the red points move closer to the center while in (c) they move further away (why?).  The blue points seem to consistently become more dense from (a) to (c).

The attack model is too rough.  It seems that without bounding D, we can make the model arbitrarily bad, no?  Assumption 1 alludes to this but doesn't specify what is "small"?  Also the attack model is described without considering if the adversary knows the learner's algorithm.  Even if there is randomness, can the adversary take actions that account for that randomness?

Does selecting a projection based on compactness remove the randomness?

Experiments -- why/how would you have distorted test data?  Making an anomaly seem normal by distorting it is easy.

I don't see experiments comparing having random projections and not.  This seems to be the fundamental question -- do random projects help in the train_D | test_C case?

Experiments don't vary the attack much to understand how robust the method is.

---

### Official Review · AnonReviewer2 · 2017-12-01
**Somewhat interesting, but lack of evidence to support statements**

**Rating:** 4
**Confidence:** 4

**Review:**

Although the problem addressed in the paper seems interesting, but there lacks of evidence to support some of the arguments that the authors make. And the paper does not contribute novelty to representation learning, therefore, it is not a good fit for the conference. Detailed critiques are as following:
1. The idea proposed by the authors seems too quite simple. It is just performing random projections for 1000 times and choose the set of projection parameters that results in the highest compactness as the dimensionality reduction model parameter before one-class SVM.
2. It says in the experiments part that the authors have used 3 different S_{attack} values, but they only present results for S_{attack} = 0.5. It would be nicer if they include results for all S_{attack} values that they have used in their experiments, which would also give the reader insights on how the anomaly detection performance degrades when the S_attack value change.
3. The paper claims that the nonlinear random projection is a defence against adversary due to the randomness, but there is no results in the paper proving that other non-random projections are susceptible to adversary that is designed to target that projection mechanism and nonlinear random projection is able to get away with that. And PCA as a non-random projection would a nice baseline to compare against.
4. The paper seems to misuse the term “False positive rate” as the y label of figure 3(d/e/f). The definition of false positive rate is FP/(FP+TN), so if the FPR=1 it means that all negative samples are labeled as positive. So it is surprising to see FPR=1 in Figure 3(d) when feature dimension=784 while the f1 score is still high in Figure 3(a). From what I understand, the paper means to present the percentage of adversarial examples that are misclassified instead of all the anomaly examples that get misclassified. The paper should come up with a better term for that evaluation.
5. The conclusion, that robustness of the learned model increases wrt the integrity attacks increases when the projection dimension becomes lower, cannot be drawn from Figure 3(d). Need more experiment on more dimensionality to prove that.
6. In the appendix B results part, sometimes the word ’S_attack’ is typed wrong. And the values in  “distorted/distorted” columns in Table 5 do not match up with the ones in Figure 3(c).

---

### Decision · Program_Chairs · 2018-01-29
**ICLR 2018 Conference Acceptance Decision**

**Decision:**

Reject

**Comment:**

The reviewers have unanimously expressed concerns about clarity, novelty, sound theoretical justification and intuitive motivation of the proposed approach.